# Ready for School: A Multi-Dimensional Approach to School Readiness Assessment in Hispanic Children from Puerto Rico

**DOI:** 10.3390/bs15070957

**Published:** 2025-07-15

**Authors:** Mary Rodríguez-Rabassa, Kamalich Muniz-Rodriguez, Allison A. Appleton, Marilyn Borges-Rodríguez, Nicole E. Ruiz-Raíces, Francisco J. Reyes-Santiago, Odette Olivieri-Ramos, Luisa I. Alvarado-Domenech

**Affiliations:** 1Clinical Psychology Program, Ponce Health Sciences University, Ponce 00716, Puerto Rico; nruiz17@stu.psm.edu; 2Department of Pediatrics, Ponce Health Sciences University, Ponce 00716, Puerto Rico; oolivieri@psm.edu (O.O.-R.); lalvarado@psm.edu (L.I.A.-D.); 3RCMI Center for Research Resources, Ponce Health Sciences University, Ponce 00716, Puerto Rico; 4Ponce Research Institute, Ponce Health Sciences University, Ponce 00716, Puerto Rico; kmuniz@psm.edu (K.M.-R.); mborges@psm.edu (M.B.-R.); 5Department of Epidemiology and Biostatistics, College of Integrated Health Sciences, University at Albany, State University of New York, Rensselaer, NY 12144, USA; aappleton@albany.edu; 6Speech-Language Pathology Master’s Program, Carlos Albizu University, San Juan 00901, Puerto Rico; freyes4@albizu.edu

**Keywords:** school readiness, early education, school readiness index, early childhood development, developmental assessment

## Abstract

School readiness during early childhood is crucial for future academic success. Existing guidelines recommend a comprehensive approach. This concurrent validation study developed a School Readiness Index (SRI) with five readiness domains: early learning skills, approach to learning, cognitive skills, socioemotional development, and physical health. Through a cross-sectional comparative design, the school readiness skills of 119 Puerto Rican children (63 males, 56 females) aged 54–65 months were assessed using standardized tests (e.g., Batería IV Woodcock-Muñoz and NIH Toolbox Cognition Battery), parental questionnaires (e.g., Ages and Stages Questionnaire-3), and physical health assessments. Each measure was scored and classified using a binary coding system (0 and 1) based on participant abilities (e.g., 1 for expected performance, 0 if below expectations). A composite SRI score was calculated using 25 indicators. Discriminant validity was assessed by comparing children’s registration status in the special education program (SEP). Sex, household income, and maternal education are key determinants of school readiness. Children registered in the SEP had significantly lower composite scores than those not registered, supporting the SRI’s discriminant validity. The SRI is a reliable tool for identifying Hispanic children from Puerto Rico who may benefit from additional support. Inclusive and multidisciplinary assessment strategies are essential.

## 1. Introduction

The dynamic interplay between children, families, and schools determines how children use their abilities, skills, and attitudes for a seamless and successful transition to formal education, defined as school readiness ([49]). For the child, school readiness encompasses physical and emotional well-being, social and behavioral development, cognitive skills, and an effective approach to learning ([49]). Assessment practices should consider these multifaceted aspects while exploring aspects beyond academic performance ([12]). School readiness predicts lifelong physical well-being, general knowledge, socioemotional well-being, and academic success in emerging adulthood ([19]). Notably, early literacy development has been associated with improved health outcomes. Individuals with higher literacy rates tend to exhibit better physical health, engage in healthier behaviors, have lower cardiometabolic risk, and experience lower levels of anxiety and depression ([5]; [42]). Therefore, readiness must be addressed as “a health-relevant, measurable outcome” with potential long-term health consequences at the individual and population levels ([33]). Most importantly, readiness assessment work-in-progress by educators is prioritizing measurement challenges as an opportunity to mobilize investment and support for children from diverse backgrounds ([34]). Education leaders in Puerto Rico, an unincorporated state of the U.S., share similar challenges and opportunities. Readiness assessment represents a pathway to providing early intervention and special education services, as mandated by the Individuals with Disabilities Education ACT (IDEA) ([32]).

A fundamental guidance for assessing children’s school readiness is based on the National Education Goals Panel’s 1991 developmental trajectories and domains ([38]; [49]). The domains include “physical well-being and sensory-motor development”, “social and emotional development” (e.g., self-regulation, attention, and impulse control), “language development” (e.g., listening, speaking, and literacy skills), “general knowledge and cognition” (e.g., early literacy), and “approaches to learning” (e.g., enthusiasm, curiosity, and culture) ([49]). These domains lay the foundation for selecting and developing valid measures of children’s school readiness that identify strengths and needs as predictors of future success and in the context of social determinants of health ([16]; [25], [24]; [31]). The 2016 National Survey of Children’s Health (NSCH) analyzed 18 items in four school readiness domains, showing that 42.2% of preschoolers (aged 3–5) were “on track” or “healthy and ready to learn” ([25]). Of note, factors such as special health care needs status/type, screen time, adverse childhood experiences, and neighborhood amenities influenced children’s readiness. A follow-up study using 2022 NSCH data analyzed 27 items for five school readiness domains ([24]), showing an increase to 63.6% of preschool children “healthy and ready to learn.” These findings support the influence of previously mentioned factors on children’s readiness and identified others, such as male sex and parental education. It is worth noting that although valid, the NSCH data is based on parental reports in the absence of professional health and education perspectives.

Other school readiness assessments combine multiple tools to gather data from parents, teachers, and children’s test scores. The Early Development Instrument (EDI), a teacher-completed measure of school readiness for Canadian children, assesses five domains using 103 items from existing instruments, key informant interviews (educators and early childhood experts), and focus groups ([33]). The EDI has “acceptable psychometric properties” as a population-level measure of school readiness. Notably, and consistent with other studies ([7]; [31]), male children scored significantly lower than females in all domains. However, a study exploring three school-readiness domains (school-entry academic, attention, and socioemotional skills) among six longitudinal data sets as predictors of later achievement ([16]) found no significant differences due to sex. Interestingly, unlike other studies linking disparities to household income ([26]; [27]), the authors found no significant differences in school readiness based on socioeconomic status. Despite varying measurements and data collection methods (e.g., parents, teachers, school administrators, home visits, and test scores), a meta-analysis determined that school-entry math, reading, and attention skills were the strongest predictors of later achievement, while socioemotional skills were not significant.

Although school readiness has been extensively studied in general U.S. populations and Canada, research on its application to Puerto Rican children is limited. Puerto Rico faces unique challenges, including high rates of childhood poverty (54%), limited access to early childhood education programs ([45]), and the impact of natural disasters and public health emergencies. These factors may affect early development and school readiness. Despite these adversities, empirical research on school readiness in Puerto Rican children is scarce. This study addresses the gap by integrating a multidisciplinary team of health professionals into the school readiness assessment process, enhancing traditional approaches that rely primarily on parent and teacher reports. This approach is particularly valuable for vulnerable populations such as participants of this study, children in the Pediatric Outcomes of Prenatal Zika Exposure (POPZE) cohort study ([1]).

POPZE assesses the developmental skills of Spanish-speaking children born during the Puerto Rico 2016–2017 Zika epidemic with and without the prenatal exposure. Children aged 54–65 months were assessed for school readiness skills guided by the theoretical framework of the American Academy of Pediatrics (AAP) technical report on school readiness and aligning with the school readiness developmental domains identified by the National Education Goals Panel ([38]; [49]). Data to inform five school readiness domains was collected from a collaborative, multidisciplinary team comprising parents, pediatricians, psychologists, a pediatric audiologist, and other specialists in human behavior.

The main objective of this study is to contribute to the understanding of school readiness in Hispanic Puerto Rican children by developing and validating a multidimensional School Readiness Index (SRI). We describe the methodology used to construct the index and examine its discriminant validity. The team hypothesizes that children with special needs will score significantly lower than children without, and significant differences will be found based on child sex, household income, and maternal education. The inclusion of Puerto Rican children, historically underrepresented in research, ensures scientific and public health relevance, contributing novel evidence to inform education, pediatric, and policy initiatives for equitable school readiness.

## 2. Materials and Methods

### 2.1. Participants

Study participants include children from the POPZE cohort study, which enrolled children born during the Zika epidemic and subsequent post-epidemic periods in Puerto Rico between May 2016 and August 2018 ([1]). The current study focused on assessing the children’s school readiness skills when they reached 54 to 65 months. A total of 119 participants were enrolled in this study, comprising 63 males and 56 females, with an average age of 57.69 ± 3.73 months. The maternal average age was 32.26 ± 5.93 years. Additional sociodemographic characteristics are presented in Table 1. Notably, most mothers were married or cohabitating (74.8%), and nearly half (49.6%) were employed. Furthermore, 48.7% of households had an annual income of $15,000 or more. In addition, most participants had public health insurance (73.9%). Parents also reported their child’s registration status in the Special Education Program of the Puerto Rico Department of Education. It is worth noting that 32.8% of the children were enrolled under the provisions of the Individuals with Disability Educational Act ([32]). This study underwent review and approval by the Ponce Medical School Foundation Institutional Review Board (IRB approval number 1903008631; 20 May 2019). Parents provided written informed consent for their participation and their child’s participation in this study.

### 2.2. Assessment Procedures

The team conducted a comprehensive assessment to compute the SRI. The participants’ assessments were individually performed over two in-person visits. During the first visit, in a timeframe of approximately three (3) hours, trained study personnel screened the child’s vision and assessed their motor skills. Advanced clinical psychology doctoral students, supervised by an experienced licensed clinical psychologist, administered the psychometric tests to assess academic and cognitive skills. Parents or legal guardians completed questionnaires on the child’s development. The visit concluded with a comprehensive physical and neurological evaluation by the study pediatrician, with findings recorded in the Child Investigation Form (CIF), a structured form developed by the study team. A shorter second visit was for the child’s audiological evaluation by a licensed pediatric audiologist.

### 2.3. Measures

The team used valid and standardized tests and questionnaires in Spanish (Figure 1; Appendix A) to assess specific developmental areas of cognition, language, motor skills, socioemotional skills, psychological health, sensory function, basic learning skills, and the home environment’s role in fostering child development. The administration and scoring procedures followed the guidelines outlined in the relevant technical manuals. The clinical psychologist principal investigator scored and interpreted each instrument administered. Results from each measure were collected and entered in the REDCap 13.1.27 ([29], [28]) secure web application hosted by the Ponce Medical School Foundation dba. Ponce Research Institute.

#### 2.3.1. Batería IV Woodcock-Muñoz: Pruebas de Aprovechamiento (Brief Academic Battery)

The Woodcock-Muñoz Brief Academic Battery (W-M, α = 0.96–0.97) ([47]; [50]) assesses brief academic skills such as reading, writing, and math. Administered by trained/advanced psychology students, the Brief Academic Battery includes letter and word identification, spelling, and applied problems subtests. For each area, total scores are transformed to age-equivalent performance (AE) following established test norms.

#### 2.3.2. Home Observation Measurement of the Environment—Short Form (HOME-SF)

The HOME-SF (α = 0.66–0.90) ([6]; [37]) assesses parental activities with the child stimulating the child’s cognition (6 items) and providing emotional support (5 items). The team revised the question “Does the child have the use of a CD player, tape deck, or tape recorder, or record player at home and at least five records or tapes?” to reflect current media access: “Does the child have access to a smartphone, tablet, videogames, or computer that allows him/her to listen to music, watch videos or movies, and play games?” Scoring procedures for screen time (television) were revised based on the American Academy of Pediatrics guidelines for children’s digital media use ([11]). Instead of four hours as adequate, the new limit was set to one hour. Higher scores indicate a higher level of parental stimulation of a child’s cognitive development and emotional support.

#### 2.3.3. NIH Toolbox Early Childhood Cognition Battery (NIHTB-Cog)

The NIHTB-Cog ([13]) is a web-based test used to evaluate a child’s cognitive abilities. The Spanish version of the NIHTB-Cog (intra-class correlation coefficient (ICC) = 0.76–0.96) has been validated to assess attention and executive functions, episodic memory, and language ([3]; [23]; [51]). Administered by trained/advanced psychology students through the NIH Toolbox iPad app, the NIHTB-Cog has been validated in the Hispanic population ([9]; [21]) and generates age-corrected norm-based standard scores. A score below 85 indicates performance at least one standard deviation below the norm.

#### 2.3.4. NIH Toolbox Parent Proxy Emotion Battery (NIHTB-EM)

The NIHTB-EM (α = 0.66–0.82) ([41]) assesses parental perception of their child’s psychological well-being (positive peer interaction), social relationships (positive affect and empathic behavior), and negative affect (anger and anxiety). These Likert-type scales measure how often or true the child shows these behaviors. Parental reports are entered into the NIH Toolbox iPad app, where scores are transformed into t-scores. For negative behaviors (i.e., anger and anxiety), a t-score ≥60 implies at least one standard deviation above the norm. For positive socioemotional skills (e.g., positive peer interaction, positive affect, and empathic behavior), a t-score ≤40 represents at least a standard deviation below the norm.

#### 2.3.5. Ages and Stages Questionnaire, Third Edition (ASQ-3)

The ASQ-3 ([44]) assesses parental perception of a child’s development across various domains, including communication, gross motor, fine motor, problem-solving, and personal-social domains. A score at or below the age-based cutoff score indicates a potential risk of developmental delay.

#### 2.3.6. Ages and Stages Questionnaire: Social-Emotional, Second Edition (ASQ:SE-2)

The ASQ:SE-2 (α = 0.90) ([43]) assesses parental perception of a child’s socioemotional responses. A score above the age-specific cutoff score (70) indicates a potential risk for socioemotional difficulties in the child.

#### 2.3.7. Peabody Developmental Motor Scale, Second Edition (PDMS-2)

The PDMS-2 (α = 0.96–0.98) ([20]) assesses fine and gross motor skills. Of note, since the PDMS-2 instructions are not available in Spanish, the team standardized its administration. Trained study personnel adhered to the PDMS-2 age-appropriate items, scoring, and classification guidelines. A cutoff total score below 90 in the total motor quotient indicates that a child’s motor performance is at least one standard deviation below the norm.

#### 2.3.8. Instrument-Based Vision Screening

The instrument-based vision screening is conducted using the Welch Allyn Spot Vision Screener ([4]; [14]). This device assesses the vision in both eyes to identify asymmetry, such as in amblyopia, and other potential risks for vision problems that warrant further evaluation. The sensitivity for detecting amblyopia risk factors is 89.5%, and the specificity is 76.7% ([40]). Trained study personnel position the child three feet from the device and ask them to focus on its light and sounds. During this period, the device captures measurements, including refractive error, eye alignment, and other parameters. This non-invasive instrument records the data and provides information on abnormal or asymmetric visual acuity, as well as risks for myopia, hyperopia, and astigmatism. The results from the instrument are categorized as either “Pass”, indicating no referral is required, or “Fail/Refer”, indicating the need for additional evaluation.

#### 2.3.9. Audiological Evaluation

The audiological evaluation is conducted in person by a licensed pediatric audiologist in accordance with the American Academy of Audiology guidelines ([2]). A detailed interview obtains information about past medical history, birth history related to hearing, specific otologic history, exposure to loud noises, and any past symptoms that may be indicative of hearing difficulties. The audiological screening components included otoscopy, distortion product otoacoustic emissions (DPOAEs) using the Sentiero Advance from Path Medical ([39]), pure-tone or warble-tone screening using the Maico MA 25 audiometer ([36]; [39]), and automated auditory brainstem response (ABR) testing using the easyScreen by Maico ([35]). A detailed description of each component and its corresponding pass criteria is provided in Appendix A.

#### 2.3.10. Pediatrician Assessment (POPZE Child Investigation Form; CIF)

The pediatrician assessment entails a comprehensive, standardized medical history and examination of a child’s physical and neurological health. The study pediatrician conducts the in-person assessment at the study clinic and collects the child’s past health history, developmental milestones, and recent health/disease events (i.e., nutrition, temperament, sleep, preventive and acute health care visits, hospital admission, surgery, education, and specialized services) from the parent/caregiver. Data is recorded in the Child Investigation Form, a standardized age-appropriate form developed by the study team.

Upon the completion of the evaluation, two pediatricians from the research team assess the relevant physical health outcomes that determine whether the child is classified as “Healthy” or “Not Healthy.” The clinical impression of the pediatricians is based on a comprehensive review of the medical history for evidence of severe or chronic conditions, hospitalizations, or surgeries; anthropometric measures to assess obesity status ([10]); and the presence of significant abnormal physical or neurologic findings that could impair functioning. A detailed description of each clinical health criterion used to assess internal consistency is provided in Appendix A.

### 2.4. Defining the SRI

The SRI is grounded in the integration of five evidence-based developmental domains: (1) Early Learning skills (EL), (2) Approach to Learning (AL), (3) Cognitive Skills (CS), (4) Socioemotional development (SE), and (5) Physical Health (PH) (Figure 1; Appendix A). EL evaluates early academic skills and questions regarding the child’s verbal abilities. AL evaluates attitudes and motivation within the home environment regarding the child’s cognitive stimulation and emotional support. CS assesses problem-solving, memory, attention, executive functions, and language abilities. SE assesses social skills, emotional regulation, and interpersonal interactions. PH evaluates physical and neurological examination, motor skills, and sensory screening.

#### 2.4.1. Scoring the SRI

The initial calculation of the SRI was based on results derived from tests and procedures conducted in alignment with established assessment guidelines. These results were evaluated against pre-defined cutoff points to determine whether the child’s performance was as expected (“On track”) or below expectations (“At risk”).

To ensure standardization, each assessed variable was operationalized into a binary outcome, categorizing the child’s performance as either “On track” or “At risk” (See Appendix A). Specifically, the skills measured were scored using a binary system: a score of 1 indicated the presence of the skill or attribute, classifying the child as “On track”, whereas a score of 0 indicated its absence, classifying the child as “At risk.” For each assessment instrument, the thresholds used to determine whether a child was “On track” or “At risk” were based on established benchmarks for risk or developmental delay ([13]; [20]; [41]; [43], [44]). In cases where no established benchmarks were available, the lower tertile of the score distribution was used to define risk (e.g., most extreme tertile = “At risk”, rest of the distribution “On track”). We elected to use tertiles over more extreme cut points (e.g., quartiles, quintiles) to preserve statistical power while still retaining focus on the risk end of the distribution.

The SRI integrates results from ten different assessment instruments. The binary scoring system simplifies the process of simultaneously evaluating multiple aspects of school readiness. Each domain of school readiness was assigned a total score, ranging from 2 to 6, contingent on the number of skills assessed. A summative approach was employed to compute an overall composite score. Binary scores from all five domains were aggregated to produce an unweighted total score, with a maximum possible score of 25 points. The following sections provide a detailed description of the scoring procedures specific to each domain.

##### Early Learning Skills (EL) Domain

The EL domain encompasses six areas of interest (Appendix A). Three of these areas are assessed through the W-M letter and word identification, spelling, and applied problems subtests ([47]; [50]). For each area, the team evaluates the difference between the child’s age-equivalent (AE) performance and their chronological age (CA). This comparison determines whether the participant is “On track” (1 point) or “At risk” (0 points). Specifically, a participant is considered “On track” if their AE performance aligns with or surpasses their CA. Conversely, they are classified as “At risk” if their AE performance lags their CA by 12 months or more (AE − CA ≤ −12). For example, if a 60-month-old child demonstrates AE performance equivalent to that of a 45-month-old child in a particular area (45 − 60 = −15), the child is categorized as “At risk.” Conversely, if the same child demonstrated AE performance equivalent to a 49-month child (49 − 60 = −11), the child is considered “On track” since the difference is less than −12. Notably, performance that aligns with the child’s CA (60 − 60 = 0) or exceeds, such as that of a 65-month child (65 − 60 = 5), warrants an “On track” classification. This scoring system clearly delineates a number line, designating any deviation −12 or more to the left as “At risk” and positions to the right as “On track.”

The EL domain encompasses three additional areas of interest pertaining to the participant’s verbal expression, considering parental responses on standardized questions on the children’s ability to (1) be understood by others, (2) express themselves, and (3) organize their ideas. This information was derived from the Clinical Assessment documented in the Child Investigation Form. For each of these abilities, a score of 1 is assigned if the participant demonstrates the ability; otherwise, a score of 0 is assigned. A total EL score is determined by summing the scores from the W-M subtests and the verbal expression questions, with a maximum possible score of six (6) points.

##### Approach to Learning (AL) Domain

This domain assesses the home environment using the HOME-SF ([6]; [37]) questions, which evaluate cognitive stimulation and emotional support (Appendix A). If a participant’s total score for cognitive stimulation falls within the first tertile, they are classified as “At risk” (0 points); otherwise, they are classified as “On track” (1 point). Similarly, if a participant’s total score for emotional support falls within the first tertile, they are also classified as “At risk” (0 points); otherwise, they are classified as “On track” (1 point). A total AL score is calculated by summing the cognitive stimulation and emotional support scores, with a maximum possible score of two (2) points.

##### Cognitive Skills (CS) Domain

The CS domain evaluates attention and executive functions, episodic memory, and language using the NIHTB-Cog ([3]; [23]; [51]). The standard score for each area is classified as “On track” or “At risk” based on a cutoff point of 86 (Appendix A). Participants with standard scores above 86 are considered “On track” (1 point), while those below the cutoff score are considered “At risk” (0 points). The domain also assesses language development using the ASQ-3 Communication domain ([44]). Participants on or above the age-specific cutoff score are considered “On track” (1 point), while those below the cutoff score are considered “At risk” (0 points). The points evaluated in the CS domain are added for a total score of five (5) points.

##### Socioemotional Development (SE) Domain

The SE assesses the socioemotional skills of the participants using the ASQ:SE-2 ([43]) and the NIHTB-EM ([41]) (Appendix A). For the ASQ:SE-2, if the total score exceeds the age-specific cutoff score (70), it is considered an “At risk” skill (0 points); otherwise, it is classified “On track” (1 point). Similarly, for the NIHTB-EM, t-scores on positive socioemotional skills (i.e., positive peer interaction, positive affect, and empathic behavior) are categorized based on a cutoff of 40. If the evaluated results are at or above the cutoff, they are considered “On track” (1 point). Conversely, if the total t-score is below 40, it is classified “At risk” skill (0 points). A cutoff t-score of 60 is used on measures of negative affect (i.e., anger and anxiety) to classify the participant’s skills as “On track” (less than 60 = 1 point) or “At risk” (at or above 60 = 0 points). The areas evaluated by the SE domain are summed up to obtain a maximum total score of 6 points.

##### Physical Health (PH) Domain

This domain encompasses a standardized pediatrician assessment that evaluates the participant’s health status. The assessment categorizes participants into two groups: “Healthy” (1 point) or “Not Healthy” (0 points). To enhance the assessment, the PH also incorporates fine and gross motor developmental outcomes from the ASQ-3 ([44]) and the PDMS-2 ([20]) results. For ASQ-3 gross motor, results exceeding the age-specific cutoff score are designated as “On track” (1 point), while those below the cutoff score are considered “At risk” (0 points). Similarly, the total score for ASQ 3 fine motor skills is categorized as “On track” (1 point) or “At risk” (0 points) based on the age-specific cutoff score. For the PDMS-2, total motor quotient scores of 90 or higher are classified as “On track” (1 point), while those below 90 are classified as “At risk” (0 points).

The domain encompasses results from vision and audiological screenings to identify sensory deficits or potential risk factors that may impact participants’ learning readiness. For each screening test, participants who obtain a “pass” result are classified as “On track” (1 point); otherwise, they are classified as “At risk” (0 points). An overall score for the PH domain is computed from each measured area, resulting in six (6) possible points (Appendix A).

#### 2.4.2. Interpretation of the SRI

The composite score can be interpreted as a continuous variable, where higher scores indicate greater school readiness. However, the developmental domain and individual skill approach offer the most effective opportunity for individualized analysis, which can benefit children by identifying their strengths and areas for improvement in school readiness.

### 2.5. Data Analyses

Descriptive statistics were calculated for the sociodemographic characteristics of the study population. Categorical variables were analyzed using descriptive analyses, including frequencies and percentages, while continuous variables were described by measures of central tendency. Pearson correlation tests examined the bivariate associations among all SRI domains and composite score. Then, a two-step analysis examined the associations between sociodemographic characteristics and the composite score. The initial step involved a simple linear regression for each sociodemographic variable and its association with the composite score. The subsequent step involved a multivariable linear regression to test the adjusted associations between household annual income, maternal education, child’s sex, and registration status in the Special Education Program with the composite score. Household annual income was dichotomized as $14,999 or less versus $15,000 or higher. Maternal education was dichotomized as certificate, high school or less versus associate degree or more. Analyses were conducted with an alpha level of 0.05. Data management and data analysis were conducted with IBM SPSS Statistics 28.0.0.0.

## 3. Results

### 3.1. SRI Developmental Domains and Composite Scores

The SRI tool evaluated five developmental domains in Hispanic Puerto Rican children enrolled in the POPZE study: Early Learning skills (EL), Approach to Learning (AL), Cognitive Skills (CS), Socioemotional development (SE), and Physical Health (PH). When assessing the overall sample, the mean score for EL was 3.7 ± 1.6 (out of 6 points) (Appendix A). The mean score for AL was 1.3 ± 0.8 out of a maximum of 2 points. The average score for CS was 3.8 ± 1.4 (out of 5 points). The SE average score was 4.3 ± 1.5 (out of 6 points), and the PH mean score was 4.8 ± 1.3 (out of 6 points). The average composite score for the overall sample was 18.0 ± 4.5 out of 25 points, with 75% of participants having less or equal to 21 total points. Of interest, when examining specific elements within the EL domain, 64.7% of parents responded that the child was able to organize and express ideas, and 81.5% reported their child could retell a story in a logical order (Appendix A).

### 3.2. Relationships Among the SRI Composite Scores, Domains, and Determinants of Health

Pearson correlation analysis and univariate and multivariable linear regressions were performed to discern the unique and collective contributions of various factors to the variation observed in the domain-specific scores. As expected, the SRI composite score was positively correlated with its component domains; correlations ranged from (*r* = 0.46 to *r* = 0.77, all *p* < 0.01; Appendix A). Moderate positive correlations were also observed between each SRI domain, ranging from *r* = 0.20 to *r* = 0.60; *p* ≤ 0.05. Marginally significant correlations were observed for SE with EL domains (*r* = 0.15, *p* = 0.096) and PH with AL domains (*r* = 0.17, *p* = 0.08). These significant but moderate correlations suggest that each domain is related to the others yet contributes unique and additional information to the overall composite SRI score, supporting the construct validity of the SRI to measure the school readiness construct in Hispanic Puerto Rican children effectively.

Univariate models revealed statistically significant relationships between the SRI composite score and the participant’s sex, annual household income, and whether the participant was registered in the Special Education Program (SEP) of the Puerto Rico Department of Education (Table 2). Female children had significantly higher composite scores than males (B = −1.89, SE = 0.80, *p* = 0.02). Children of higher household annual income had significantly higher SRI composite scores than their counterparts (B = 1.73, SE = 0.81, *p* = 0.03). Also, children registered in the SEP had significantly lower SRI scores (B = 4.47, SE = 0.77, *p* < 0.001). These results suggest that the child’s performance in the SRI may be influenced by the child’s sex, socioeconomic status, and SEP registration status.

The results of the multivariate regression models exploring associations between SRI composite score and SEP registration status, adjusted by child sex and household annual income, are listed in Table 3. The final model included significant predictors determined by a stepwise selection procedure. The model reveals that SEP registration status was significantly associated with participant’s SRI composite score (B = 4.17, SE = 0.77, *p* < 0.001). The model also showed that household annual income (B = 1.34, SE = 0.71, *p* = 0.06) and sex (B = −1.16, SE = 0.81, *p* = 0.11) were no longer significantly associated with the SRI score. Based on the findings, participants who were not registered in SEP had, on average, a 4.17 unit increase in SRI scores compared to those registered, after adjusting for predictors. These findings suggest that SEP status is a more significant determinant of the child’s composite score, supporting the SRI’s ability to differentiate among children with and without support service needs within this population.

## 4. Discussion

This study presents a valid and comprehensive tool to evaluate school readiness in Hispanic Puerto Rican children at entry level up to five years of age (65 months) that combines multisource validated psychological and psychometric instruments, parent questionnaires, and clinical provider assessments. The SRI-based assessment provides a holistic approach that is helpful for parents, educators, and health providers in identifying a child’s school readiness strengths and needs and providing them with the necessary support to ensure a successful transition into the formal academic experience. It considers five school readiness domains assessed independently: early learning skills, approach to learning, cognitive skills, socioemotional development, and physical health. Twenty-five skills/attributes are captured among these domains, allowing for a more thorough understanding of the child’s abilities, while highlighting areas that may require strengthening for school success.

Consistent with our study hypotheses, the SRI composite score was significantly associated with factors linked to disparities in early development. In univariate analyses, female sex and higher household income were associated with higher SRI scores, while children enrolled in the Special Education Program had significantly lower scores. However, in multivariate models, only Special Education Program registration remained a significant predictor of SRI performance after adjusting for sex and income. Although maternal education did not reach statistical significance, a trend toward an association was observed (*p* = 0.06), suggesting a potential influence that may warrant exploration in further studies. These findings support the discriminant validity of the SRI and its utility in distinguishing between children with and without identified support needs, reinforcing its potential value in education and clinical decision making.

The SRI assessment can guide decision making in systems responsible for the health and developmental needs of children across Puerto Rico and in other underserved populations with similar sociodemographic and environmental risk profiles. A composite score of up to 25 points provides a screenshot of the study population’s overall performance, enabling the identification of the proportion of children who require assistance or intervention. The analyses of POPZE participants’ outcomes revealed an overall mean score of 18 points, suggesting that these children might require additional support and closer mentoring to maximize their potential. Furthermore, domain-specific analysis might facilitate resource allocations based on areas of greatest need. For instance, the POPZE children encountered significant challenges in the early learning domain. Consequently, families, educational systems, and clinical providers should prioritize efforts to address early literacy and pre-academic skills.

Findings from regression analyses support the validity of the SRI by differentiating between children enrolled in the Special Education Program and those not registered. This finding highlights the utility of the composite score and domain-specific outcomes for early screening and identifying children who require support. The observed correlation among the domains suggests that each domain contributes unique and additional information to the overall index, supporting the SRI construct validity. Consequently, combining the domains provides a more comprehensive understanding compared to examining them in isolation. This comprehensive approach enables a richer interpretation of the collective impact of various domains on the overall construct.

Most states have adopted school readiness assessments at kindergarten entry to inform instruction, while a few states include intervention, individualized plans, and remediation to improve children’s readiness skills ([22]). In this role, the centralization of Special Education Programs is critical to ensure “fair, equitable, and high-quality education and services” for children including those with disabilities ([32]). However, early identification of children’s needs is often challenging; therefore, all children could benefit from school readiness assessments for opportunities to thrive.

Many parents are unable to recognize early learning difficulties in their children, and these challenges often become apparent at higher grade levels, when academic demands increase. School readiness assessments help identify areas of need at early life stages to channel timely interventions for better outcomes ([38]). The purpose of the SRI assessment is not diagnostic but seeks to offer recommendations that benefit Puerto Rican children. Disclosing results and actively involving parents in the process is key to fostering understanding, collaboration and ensuring tailored support for each child’s development, and attainment of school readiness.

It is vital to recognize that children’s functioning is not static. Given the dynamic nature of child development and brain neuroplasticity, early interventions open the door to significant improvements in children’s functioning. Early investment in supporting school readiness skill development can result in better life-course physical and psychosocial health and employment opportunities ([15]). However, social determinants of health, such as gender, maternal education, and household income, need to be considered. Consistent with previous findings of preschool children ([7]; [24]; [31]; [33]), female participants in this study demonstrated better school readiness competencies than males. Children with higher household income and maternal education had better school readiness scores than their counterparts, as identified in other studies ([24]; [26]; [27]), although others found no significant differences related to income ([16]). Individuals with higher education tend to have higher incomes, which can lead to better developmental outcomes for preschool children ([46]). Families with socioeconomic advantages can provide enrichment experiences to children and have access to resources, including high-quality educational or childcare centers, better nutrition, and materials and equipment that stimulate the optimum development of children. Given that the prevalence of special health care needs is disproportionately higher in children from poor and disadvantaged families ([30]), social policies are warranted to buffer and reduce disparities.

School readiness not only impacts academic success but also shapes the child’s life course. The SRI assessment includes critical skills for setting a strong foundation for future educational success ([16]; [17]; [19]; [38]). The SRI composite score stands out as a comprehensive tool that distinguishes itself from previously developed indices ([16]; [25], [24]; [33]). It accomplishes this by integrating assessments from both clinical psychology professionals and pediatric clinical providers, synthesizing their collective expertise into a unified measure. The active involvement of health professionals, especially pediatricians, becomes crucial in this narrative. Recognizing brain neuroplasticity, they can be catalysts for early interventions that benefit children, establishing a path towards equality from the beginning.

This study has several strengths. Notably, this study excels in its multidisciplinary approach and the seamless integration of professional assessments. Despite the lack of specific norms for certain tests, the availability of assessment tools in both English and Spanish allows comparisons with other populations. Also, individual skill-focused outcomes benefit children by providing the opportunity to identify strengths and needs for school readiness. The SRI’s binary scoring system, which has been implemented in other indexes, enhances accessibility, making the interpretation of results straightforward and accessible to a broad range of concerned individuals and organizations, including parents and educators ([8]; [18]).

### Study Limitations

Study limitations include the use of a binary score, which can limit the comparisons across studies when the study population differs ([48]). Also, the summative approach employed by the SRI does not discern underlying factors contributing to readiness and assigns equal weight to all components, thereby potentially overlooking their contributions to the index ([8]; [18]; [48]). Future exploratory analyses will be conducted to assess possible influential factors. For example, the impact of prenatal Zika virus exposure on learning readiness is the object of a separate manuscript. Another limitation of this study is that some of the test results are based on perceptions of parents and professionals who completed the instruments. However, it is important to note that most of these instruments have been validated, which helps mitigate potential bias. Furthermore, implementing a comprehensive school readiness evaluation using an SRI-based assessment requires substantial financial resources and time, which may limit its feasibility for large-scale use. Moreover, in disadvantaged areas, the availability of multidisciplinary teams, crucial for a thorough assessment, may be limited, further restricting the SRI application.

## 5. Conclusions

The assessment of school readiness skills is crucial for the early identification of children’s strengths and needs, ultimately promoting their successful future academic and occupational performance. Developed in accordance with established domains from the literature, the SRI integrates data from validated assessment instruments administered by a multidisciplinary team, providing a holistic assessment of the child’s functioning. The findings support the discriminant validity of the SRI, showing its ability to differentiate school readiness performance among children with and without special education needs. While female sex and higher household income were initially associated with higher SRI scores, only Special Education Program registration remained a significant predictor in the adjusted model, underscoring the SRI’s utility in identifying children who require early intervention. The inclusion of health professionals in the assessment processes enriches previous methods that predominantly rely on parent and teacher reports, therefore supporting vulnerable populations.

## Figures and Tables

**Figure 1 behavsci-15-00957-f001:**
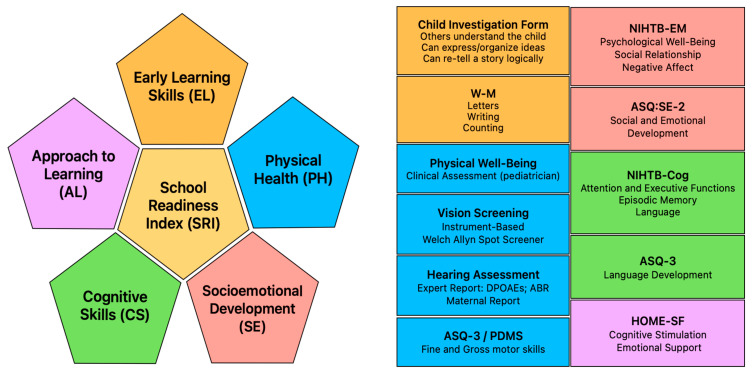
Domains of school readiness: Tools/Measures and Skill Elements Assessed. Note: Abbreviations: W-M: Batería IV Woodcock-Munoz: Pruebas de aprovechamiento; HOME-SF: Home Observation Measurement of the Environment—Short Form; NIHTB-Cog: NIH Toolbox Early Childhood Cognition Battery; ASQ-3: Ages and Stages Questionnaire-3; ASQ-SE-2: Ages and Stages Questionnaire: Social-Emotional, Second Edition; NIHTB-EM: NIH Toolbox Parent Proxy Emotion Battery; PDMS-2: Peabody Developmental Motor Scale, Second Edition; DPOAEs: Distortion product otoacoustic emissions; ABR: Auditory brainstem response screening.

**Table 1 behavsci-15-00957-t001:** Sociodemographic characteristics of study participants.

Characteristics	Values	
	*n*	%
Participant sex		
Male	63	52.9
Female	56	47.1
Special education program		
Registered	39	32.8
Not registered	80	67.2
Annual household income		
** **$14,999 or less	61	51.3
$15,000 or higher	58	48.7
Health insurance		
Public	88	73.9
Private	31	26.1
Maternal civil status		
Married or cohabitating	89	74.8
Other	30	25.2
Maternal education attainment		
Certificate, high school or less	51	42.9
Associate degree or more	68	57.1
Maternal employment status		
Employed	59	49.6
Unemployed	60	50.4

**Table 2 behavsci-15-00957-t002:** Simple linear regression results for sociodemographic variables and SRI composite score for children aged 54–65 months.

Model	Measure	Estimate	*SE*	95% CI		*p* ^e^
				*LL*	*UL*	
SLRM #1	Intercept	18.98	0.58	17.83	20.14	<0.001
	Participant sex ^a^	−1.89	0.80	−3.48	−0.30	0.02
SLRM #2	Intercept	17.09	0.62	15.87	18.31	<0.001
	Maternal education ^b^	1.56	0.82	−0.06	3.18	0.06
SLRM #3	Intercept	17.14	0.56	16.03	18.25	<0.001
	Annual household income ^c^	1.73	0.81	0.13	3.32	0.03
SLRM #4	Intercept	14.98	0.63	13.73	16.23	<0.001
	Registered in special education program ^d^	4.47	0.77	2.94	5.99	<0.001

Note. SLRM = Simple linear regression model; *SE* = standard error; CI = confidence interval; *LL* = lower limit; *UL* = upper limit. ^a^ 0 = female, 1 = male; ^b^ 0 = certificate, high school or less, 1 = Other; ^c^ 0 = $14,999 or less, 1 = $15,000 or higher; ^d^ 0 = yes, 1 = no. ^e^
*p*-value was evaluated at a 0.05 alpha.

**Table 3 behavsci-15-00957-t003:** Association between SRI composite score and special education program registration status, adjusted by annual household income and child sex.

Measure	Estimate	*SE*	95% CI		*p* ^d^
			*LL*	*UL*	
Intercept	15.13	0.85	13.46	16.81	<0.001
Participant sex ^a^	−1.16	0.72	−2.58	0.27	0.11
Annual household income ^b^	1.34	0.71	−0.08	2.75	0.06
Registration in special education program ^c^	4.18	0.77	2.66	5.69	<0.001

Note. *SE* = standard error; CI = confidence interval; *LL* = lower limit; *UL* = upper limit. ^a^ 0 = female, 1 = male; ^b^ 0 = $14,999 or less, 1 = $15,000 or higher; ^c^ 0 = yes, 1 = no. ^d^
*p*-value was evaluated at a 0.05 alpha. Constant = 15.13, F = 13.87, *p* < 0.001, R^2^ = 0.266.

## Data Availability

The data that support the findings of this study are available from the corresponding author upon reasonable request.

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
