# Peer review of "Ready for School: A Multi-Dimensional Approach to School Readiness Assessment in Hispanic Children from Puerto Rico"

_behavsci, 2025, doi:10.3390/bs15070957_

Round 1

Reviewer 1 Report

Comments and Suggestions for Authors

Abstract is concise and well written.

Introduction - I was expecting a closer examination of school readiness within Puerto Rican children. This is the biggest need of the paper- tell me why these articles and concepts cited are relevant to Puerto Ricans. Tell me how this work contributes to knowledge gaps informing Puerto Rican education and school readiness. I also suggest a more action-oriented purpose statement (lines 106-110). I believe the purpose of the work is to inform Puerto Rican school readiness and the methods are to assess discriminate validity. 

Materials and Methods - This section was very well written. I appreciated the notes about the validity of these assessments with Spanish speakers. This added credibility to the methods and overall study design. I thought the battery of tests were balanced and showed a comprehensive view of child functioning. For section 2.3.9 and 2.3.20, this detail becomes hard to grasp, and I wonder if a table would better show the details while you keep the main ideas in the text? You describe 16 different measures, and while this context is helpful, there is so much information that most readers will scan or skip. Can you have a table with details and brief descriptors in the text?

Results - Very brief and concise. I could follow the information easily even though these methods are not my area of expertise.

Discussion - Define "other populations" (line 504). Why would your results generalize? If you only looked at Hispanic Puerto Ricans I am curious why you think the results apply to non-Hispanic Puerto Ricans. Otherwise, I found your discussion section to be well written and conceptualized. I prefer to see Limitations as a separate section rather than integrated, but the journal may have other guidelines.

Conclusions- brief but warranted and fair.

Reviewer 2 Report

Comments and Suggestions for Authors

Dear author(s),

I have read with much interest your paper titled “Ready for School: A Multi-Dimensional Approach to School Readiness Assessment in Hispanic Children from Puerto Rico”.

The paper present useful data regarding the methodology used to develop the School Readiness Index (SRI) for Hispanic Puerto Rican children and examines its discriminant validity. Additionally, the study explores whether significant differences in school readiness emerge based on child sex, household income, and maternal education, aiming to establish the index’s sensitivity to key sociodemographic variables.

However, I recommend:

  1. To state more clearly how the results align (or not) with the stated hypotheses and what implications this has for the field.
  2. To highlighting the limitations of the research in a separate (sub)section.

Reviewer 3 Report

Comments and Suggestions for Authors

Manuscript summary 

The author(s) stated that the primary objective of the study is to describe the methods used to develop the School Readiness Index (SRI) for Hispanic Puerto Rican children and to test its discriminant validity. They hypothesize that children with special needs will score significantly lower than those without special needs. Furthermore, they expect to find significant differences based on child sex, household income, and maternal education levels. The author(s) argued that while the National Survey of Children's Health (NSCH) data is valid, it relies on parental reports, which excludes the perspectives of health and education professionals. They conducted a comprehensive assessment to compute the SRI.

Reviewer comments:

Title, Abstract, and Introduction – overall evaluation
Acceptable. The abstract captures the readers' attention.

Methodology / Materials and Methods – overall evaluation
Scientifically sound. The author(s) stated that the participants included the Pediatric Outcomes of Prenatal Zika Exposure (POPZE) cohort and that the Woodcock-Muñoz Brief Academic Battery was employed to assess essential academic skills in reading, writing, and math. In addition, a home observation was conducted to evaluate the home environment. The ASQ-3 was also used to gain insights into parental perceptions of their child's development across various areas, including communication, gross motor skills, fine motor skills, problem-solving, and personal-social development. These assessments were just a few of the tools utilized in the evaluation.

What I found noteworthy is that the author(s) included a framework of the SRI and how it integrates five key domains: early learning skills, approach to learning, cognitive skills, socio-emotional development, and physical health. They explained that they employed a binary system to categorize each child's performance, determining whether they were on track or at risk. A score of 1 indicates the presence of a skill or attribute, classifying the child as on track, while a score of 0 indicates its absence, classifying the child as at risk. This process aligns with the aim of the study.

Compliance with Ethical Standards – The author(s) indicated that this study was reviewed and approved by the Institutional Review Board of the Ponce Medical School Foundation (IRB approval number 127 1903008631; May 20, 2019). Parents provided written informed consent for both their participation and their child's participation in the study.

Results/Data Analysis– overall evaluation
Sound. The author(s) conducted a multi-step analysis of the data, which included descriptive statistics to assess the sociodemographic characteristics of the study population as well as Pearson correlation tests to explore the bivariate associations among all SRI domains and the composite score. Additionally, the author(s) conducted a two-step analysis to investigate the relationships between sociodemographic characteristics and the composite score. The author(s) noted a link between SRI scores and participants' sex, household income, and enrollment in the Puerto Rico Department of Education's Special Education Program (SEP). Children in the SEP had significantly lower SRI scores.

Figures/Tables – overall evaluation
Sound. Figures and tables were clear and easy to understand. The author(s) consistently and accurately interpreted the data. It was observed that the data suggested that SEP status significantly influences a child's score, highlighting the SRI's ability to differentiate between children who need support services and those who do not.

Interpretation /Discussion – overall evaluation
Minor modification. The author(s) stated that their findings support the use of the SRI by differentiating between children in the Special Education Program and those who are not enrolled. They presented a strong argument, asserting that the SRI-based assessment offers a comprehensive approach that is beneficial for various stakeholders, including parents, educators, and health providers. They claimed that this assessment can guide decision-making, help determine whether children are ready for school, identify their strengths and needs, and provide the necessary support to ensure a successful transition into a formal academic environment.

The author(s) also discussed the Pediatric Outcomes of Prenatal Zika Exposure (POPZE) assessment, which focuses on children born during the Zika epidemic. They emphasized the need for increased support and mentoring to help these children thrive, noting that many parents may not recognize early learning challenges that become apparent in higher grades. While the authors argue that the SRI assessment is not diagnostic, they believe it offers valuable recommendations for Puerto Rican children.

The discussion section aligns with a literature review, reinforces the study’s findings, and contextualizes an argument for the use of the SRI. It is recommended to include how the study’s findings clearly support the hypothesis.

Conclusions – overall evaluation
Minor modification.  The author(s) of the study aim to “describe the methods used to develop the School Readiness Index (SRI) for Hispanic Puerto Rican children and to test its discriminant validity.” They hypothesize that “children with special needs will score significantly lower than children without special needs, and that significant differences will be found based on child sex, household income, and maternal education.”

In the conclusion section, the author(s) briefly mentioned that the SRI was developed "in accordance with established domains." While this point is significant, it is suggested that the conclusion be expanded to include a section that directly addresses the study's aims and evaluates whether the hypothesis was supported.

References – overall evaluation
Sound. Cited references are mostly recent publications (within the last 5 years) and relevant.

Writing – overall evaluation
Accepted with minor modifications

Reviewer 4 Report

Comments and Suggestions for Authors

An excellent and timely paper. Well done.

The main question addressed with the research relates to student preparedness for school. This is very important as research has indicated that if a child, for example, does not have the necessary reading skills by year 3  these children a most likely to become disengaged with school and consequently become behavioural problems etc. The inclusion of the use of health professionals was a seminal factor. 

The more that can be done to prepare students for school the better prepared they will be for engaging in their studies. And again the introduction of health professionals.  It address a specific gap in the field, as it deals specifically with a specific cohort of students, located in a specific region, with the inclusion of health professionals adds another dimension to the research. Therefore, there is a strong possibility that very little research has been undertaken on this topic with this group of students in this region.

The methodology was well throughout and developed. It provided the depth and breadth of scope to analyze the focus of the research.

But no tables and figures applied to APA 7

I found the paper of interest and the focus on the use of health professionals to assist with student readiness as a strong contributing factor to the significance of the paper. It was well written and structured. With only minor challenges, for example, the use of direst quotes to highlight key points. These I felt would have simply been reflected in single quotation marks as the authors are quoting another person.
